# Molecular Insight into Gastric Cancer Invasion—Current Status and Future Directions

**DOI:** 10.3390/cancers16010054

**Published:** 2023-12-21

**Authors:** Tasuku Matsuoka, Masakazu Yashiro

**Affiliations:** Molecular Oncology and Therapeutics, Osaka Metropolitan University Graduate School of Medicine, Osaka 5458585, Japan; t22738q@omu.ac.jp

**Keywords:** invasion, metastasis, gastric cancer, epithelial–mesenchymal transition, mesenchymal–amoeboid transition, tumor microenvironment, non-coding RNAs, targeted therapy

## Abstract

**Simple Summary:**

Gastric cancer (GC) is one of the most common malignancies worldwide. Owing to the absence of specific early symptoms, most patients with GC are often diagnosed at an advanced stage. Invasion is the most important feature of GC metastasis, which causes poor mortality in patients. Recently, genomic research has critically deepened our knowledge of which gene products are dysregulated in invasive and metastatic GC. This study summarizes the advances in our current understanding of the molecular mechanism of GC invasion and metastasis. We also highlight the future directions of the invasion therapeutics of GC in clinical use.

**Abstract:**

Gastric cancer (GC) is one of the most common malignancies worldwide. There has been no efficient therapy for stage IV GC patients due to this disease’s heterogeneity and dissemination ability. Despite the rapid advancement of molecular targeted therapies, such as HER2 and immune checkpoint inhibitors, survival of GC patients is still unsatisfactory because the understanding of the mechanism of GC progression is still incomplete. Invasion is the most important feature of GC metastasis, which causes poor mortality in patients. Recently, genomic research has critically deepened our knowledge of which gene products are dysregulated in invasive GC. Furthermore, the study of the interaction of GC cells with the tumor microenvironment has emerged as a principal subject in driving invasion and metastasis. These results are expected to provide a profound knowledge of how biological molecules are implicated in GC development. This review summarizes the advances in our current understanding of the molecular mechanism of GC invasion. We also highlight the future directions of the invasion therapeutics of GC. Compared to conventional therapy using protease or molecular inhibitors alone, multi-therapy targeting invasion plasticity may seem to be an assuring direction for the progression of novel strategies.

## 1. Introduction

In East Asia, especially Japan, the incidence rate of GC is the highest [1]. Gastric cancer (GC) is a heterogeneous and multioncogenic disease that causes 7.7% of all cancer-related deaths worldwide [2], and the identification of the most efficient treatments for GC is thus a key research challenge. Previously, advanced GC has been treated with sequential lines of cytotoxic agents, beginning with a platinum and fluoropyrimidine combination in the first line. However, the median survival obtained from clinical trials for GC remains about 11 months [3]. Noteworthy, trastuzumab combined chemotherapy in human epidermal growth receptor 2 (HER-2)-positive patients successfully prolonged survival from 11 to 14 months [4]. According to this result, advanced GC is recommended to be managed with the combinations of platinum (cisplatin, oxaliplatin) and fluoropyrimidine (capecitabine, fluorouracil, S-1)—with or without trastuzumab for the first line, followed by ramucirumab plus paclitaxel (PTX) for second-line therapy, and nivolumab or irinotecan for third-line therapy in Japan. More recently, three randomized trials, KEYNOTE-062, ATTRACTION-4, and CheckMate 649, demonstrated the effectiveness of immune checkpoint inhibitors in the first-line treatment of advanced GC [3,5,6,7]. Meanwhile, the majority of anticancer agents that are currently available act against a limited number of targets, such as cell growth or death. In addition, researchers must consider that the poor survival of cancer patients implies the process of invasion and metastasis [8]

GC has been histologically categorized into two major subtypes: intestinal- and diffuse-type based on Lauren’s histopathologic classification [9]. Intestinal-type GC, indicated by cohesive tumor cells and intestinal-type glandular differentiation, is the most popular type. In contrast, diffuse-type GC, shown in about 32% of cases, comprises poorly differentiated cells that become highly invasive and metastasize to distant organs, including the liver, lung, and peritoneum [10]. The progression of GC initiates with cancer that originates within the epithelium; it invades the lamina propria and then the submucosa, and goes through multiple steps until it forms metastatic foci in distant organs. At each step, cancer cells change their morphology and properties to protect themselves from host attacks, and they can then succeed in forming metastases. The invasion process is one of the key drivers of cancer metastasis, which represents the leading cause of cancer-related death [11]. Cancer cells display various invasion approaches, such as shifting between modes of invasion in response to different stimuli, e.g., therapeutic interventions and stimulation of the surrounding microenvironment. The capacity of cancer cells to employ one or more of the invasion modes and to switch among them is termed invasion plasticity, which presents a major challenge in the treatment of cancer metastasis. Although many advances have been made in this field over the past several decades, our knowledge of the mechanisms that underlie the regulation of cancer cell invasion is insufficient.

In this review, we first provide an overview of the various modes of invasion and the plasticity that enables cancer cells to switch between different invasion modes in GC. We also summarize the current knowledge concerning signaling pathways, genetic alterations, and non-coding (nc)RNAs which govern the mechanisms of GC invasion. We then discuss (i) the current therapeutic interventions for invasion, (ii) some strategies arising from invasion plasticity that could explain the failure of some therapies, and (iii) the potential targets that could lead to more effective inhibition of GC cell invasion.

## 2. Methods

A non-systematic review was performed based on an electronic search through the medical literature using PubMed and Google Scholar. The keywords “gastric cancer”, “invasion”, “epithelial–mesenchymal transition”, “mesenchymal–amoeboid transition”, and “tumor microenvironment” were used. In addition to research articles, which were mainly searched to obtain novel findings, review articles and guidelines investigating the roles of invasion for the progression of GC from gastroenterology, oncology, and genetics were included in this review. When more than one guideline concerning the same subject was available, the most up-to-date one was selected. Only full articles in the English language published in the last 10 years were considered for further review. Great importance was also given to “clinical study” and “review” articles dealing with the topic. The exclusion criteria comprised duplicated articles and studies absent of diagnostic outcomes. Case reports, correspondence, letters, and non-human research were not included. First, the titles were screened and appropriate studies were selected. Of these studies, the full text was acquired. A total of 229 articles were identified.

## 3. Overview of Invasion and Metastasis

Cancer invasion and the metastasis process are the main indicators of tumor development and involve various mechanisms (Figure 1) [8]. Cancer cells that initiate within the gastric epithelium destroy the basement membrane (BM) and infiltrate into the lamina propria and then into the submucosa due to a repression in or complete deficiency of intercellular adhesion molecules, and an enhancement of aberrantly high motile ability that enables penetration through the stiff structural components of the surrounding stroma. Complexities of the metastatic process—the infiltration of the primary tumor into surrounding tissues and the establishment of the metastatic site—consists of several stages: transition in blood or lymph, invasion into the systemic circulation, extravasation, colonization in the second site, and the construction of clinically detectable metastasis [12]. For cancer cells, the path to forming metastatic foci is difficult. The reasons restraining the proliferation of malignant cells involve the BM and various elements of the surrounding stroma, increased interstitial pressure, hypoxia in tumor cells, and consistent revelation to immune cells. Meanwhile, some cells fight potently, reducing microenvironmental components and extending an aggressive phenotype and advanced metastatic capacity. The acquirement of invasive behavior includes stimulation of the signaling pathways regulating cytoskeletal dynamics, and the turnover of cell–matrix and cell–cell attachment. In addition, recent evidence suggests that perineural invasion (PNI) has been related to a substantial pathological process of various tumors including GCs [13]. PNI is ordinarily defined as a mechanism of cancer cell dissemination along nerves and tumor cells that can migrate from a primary tumor to distant sites along nerve tracts. Cancer invasion is an accommodative process including alteration of cell shape and initiation of cell polarity, which leads to translocation of the cell body [14]. Tumor cells can invade either collectively on retention of cell–cell adhesions or individually in the deficiency of cell–cell junctions [15]. As a consequence, tumor cells can utilize various strategies when migrating collectively or individually. As such, there are three modes of cancer invasion, namely, collective invasion, mesenchymal invasion, and amoeboid invasion.

Collective cell invasion involves compact and cohesive cell groups with two or more adjacent cells (Figure 2) [16]. The morphological feature of collectively infiltrating cancer cells varies from strands of a few cells to expansive masses that involve cells that do not attach to the extracellular matrix (ECM) and even construct luminal structures. It has been described that the bulky cell accumulation could secrete highly concentrated proteolytic factors and matrix proteases and defend inside cells from immune activity. A principal feature of collective invasion is tumor budding, which is usually detectable at the invasive front of cancers [17]. This is frequently seen in invasive cancers such as breast, prostate, and pancreatic cancer, but not often seen in GC.

Mesenchymal cell invasion is a distinctive mode of smooth muscle cells, endothelial cells, and fibroblasts (Figure 2) [18]. Malignant tumors deriving from bone marrow, connective tissue, and poorly differentiated epithelial neoplasms often reveal a mesenchymal type of invasion [19]. Tumor cells showing mesenchymal invasion histologically exhibit an elongated, spindle-like cell shape with the formation of pseudopod protrusions and filopodia [20]. Cancer cells acquired via the mesenchymal mode require proteases that actively transform the ECM and generate cell migration tracks by which cancer cells invade.

The amoeboid invasion is characterized by unique features involving high-velocity motility, roundish cell morphology, and low cell–ECM attachment as well as a lack of proteolytic activity in the adjacent ECM (Figure 2) [21]. This invasion procedure is thought to be the most effective invasion pattern. Quick plastic deformation, which is efficient for infiltrating through narrow gaps of the surrounding ECM, is shown in amoeboid cells [22]. This deformation is produced by the reformation of the cortical actin cytoskeleton, which leads to the moving cells contracting and expanding in high-velocity cycles, and replacement by changing their positions [20].

During the invasion process, cancer cells modify their invasion modes depending on different environmental situations by converting between single-cell and collective-cell invasion [23].

Epithelial–mesenchymal transformation (EMT) is a molecular mechanism characterized by the loss of polarity of epithelial cells, which disrupts cell attachment to the ECM (Figure 2) [24]. Subsequently, EMT leads to increased invasion capacity, which means acquiring a more mesenchymal phenotype [24]. When EMT is induced, cancer cells form filopodia and lamellipodia in the direction of their movement and adhere to interstitial tissue as they move. It has also been reported that malignant cells that have acquired EMT have novel abilities such as stem cell-like characteristics and immune tolerance. During EMT, the expression of intercellular adhesion molecules such as E-cadherin is suppressed in cancer cells. β-catenin, an intracellular binding protein of E-cadherin, is stabilized and translocated into the nucleus. It is thought that when this series of intracellular transduction systems (Wnt/β-catenin signal pathway) is activated in cancer cells, the expression of genes that induce EMT is increased. In addition, a pathway in which a group of signal molecules called SMADs are activated by transforming growth factor-β (TGF-β) stimulation plays an important role in EMT induction. EMT-related transcription factors include the Snail family (Snail, Slug, etc.), ZEB family (ZEB1, ZEB2), and Twist, and although they vary depending on the type of cancer, Snail, ZEB1, and Twist are expressed by strengthened TGF-β/SMAD signal stimulation. Numerous studies have stated that the acquisition of EMT characteristics is a crucial factor that enables cancer cells to prompt invasion and the metastatic process. For instance, collagen type IVα1 exhibited the prompted EMT and invasion ability of GC cells via the Hedgehog signaling pathway [25]. EMT has been revealed to be linked to the stemness of cancers [26]. Cancer stem cells (CSCs) are closely associated with the initiation, relapse, and metastasis of cancers owning to their self-renewal and heterogeneous division. Gastric CSCs acquire extensive invasive capacity and EMT properties. A previous study demonstrated that loss of RUNX3 (Runt-related transcription factor 3) in gastric epithelial cells induced EMT and an extremely enhanced expression of the Gastric CSC cell marker Lgr5 (leucine-rich repeat-containing g-protein coupled receptor 5) [27]. Similarly, the upregulation of SOX4, a member of the sex-determining gene family, increased the expression of EMT markers (Twist1, ZEB1, snail1) and stemness transcription factors (Oct4, SOX2) of GC cells, suggesting the induction of EMT and stemness [28].

Mesenchymal–amoeboid transition (MAT) is an indicator of invasive plasticity that facilitates tumor cell dissemination, as well as the assistance of tumor cells to restrain the effectiveness of medical treatment by adopting another molecular mode for cell invasion (Figure 2). MAT in malignant cells can be induced by a decrease of cell–ECM interaction, loss of ECM proteolysis, enhanced contractility, or the blockade of Rac activity [19], or indirectly promote Rho activation by engaging EphA2 [29]. Rounded-amoeboid invasion was also facilitated by restoring the levels of RhoA-suppressive regulator p27Kip1 [30]. Targeted inhibition of some proteases can switch cells from an acquired mesenchymal-type invasion to an amoeboid type and obtain their characteristic plasticity. In GC, RhoA/ROCK signaling induces MAT by blocking Rac activity [31]. When treated with specific inhibitors of Rho/ROCK in scirrhous GC, cancer cells revealed MAT whereas specific inhibitors of Rac, a mediator with inverse interaction of RhoA, repressed MAT [31]. A blockade of Rho/ROCK signaling can impair the expression of α3-integrin, increase membrane-type 1 matrix metalloproteinase (MT1-MMP), and thus facilitate mesenchymal invasion, namely amoeboid–mesenchymal transition (AMT).

## 4. Genetic Alteration in Invasion-Associated Genes

Tumorigenesis of the stomach comprises the gradual accretion of a variety of genetic and epigenetic alterations, which result in increased activation in oncogenes and reduced function in tumor suppressor genes. Genetic changes, including *KRAS*, *p53*, *PIK3CA*, *ARID1A*, and *MLL* mutations, as well as *ERBB4*, *CD44, PIK3CA*, and *C*-*MET* amplifications, are often identified in GC implying that they may have critical tumorigenic roles in gastric carcinogenesis [32,33,34,35]. *P53*, a recognized tumor suppressor, has been reported to prompt cell cycle arrest and apoptosis by stimulating downstream target genes [36]. *P53* mutations are the most familiar genetic alteration in numerous human cancers. A previous paper described that *TP53* mutation was related to lymphatic and venous invasion in advanced GC [37]. Meanwhile, accumulating evidence indicates that genetic alterations, epigenetic alterations including DNA methylation of CpG islands, and post-translational modifications of histones are also involved in the invasion of GC. Most of these genes target proteins relevant to signal transduction involving ligands, receptors, and intracellular modulators.

Employing microarray technology, various studies have attempted to find the gene expression profiling that was specialized in invasion and metastasis. A previous study using a cDNA microarray consisting of 23,040 genes detected 12 genes correlated with lymph node metastasis. Nine of the 12 genes are relatively upregulated (*DDOST, GNS, NEDD8, LOC51096, CCT3, CCT5, PPP2R1B*, and two ESTs) and three are downregulated (*UBQLN1, AIM2*, and *USP9X*) in GC patients with lymph node metastasis [38]. Similarly, a study using cDNA microarray identified several genes that are diversely expressed depending on the depth of tumor invasion. Among them, 10 genes, such as *SPARC, CEACAM6, KRT6B, THBS2, IGFBP3, TGFB3, MMP1/7/10*, and *CSPG2*, were upregulated and were closely related to invasion capacity [39]. Another paper described that *CDH17* and *APOE* were found to be correlated with the invasion depth of tumors by serial analysis of gene expression (SAGE) [40]. IQ motif-containing GTPase-activating protein 1 (IQGAP1) is known to be a negative modulator of cell–cell adhesion at adherens junctions in cancer. In diffused-type GC, mutations at the Cdc42 and Rac1 activation binding sites of *IQGAP1* were found to occur [41]. Additionally, IQGAP1 is upregulated in GC cells, and promotes cell invasion by targeting RhoC GTPase [42]. Also, IQGAP3 acts as an essential mediator of invasion and EMT through TGF-β signaling. IQGAP3 has a pivotal role in the invasion of GC cells and mediates cytoskeletal remodeling [43]. Another study demonstrated that the mRNA level of the *ZYX* gene in GC cells was superior to that in adjacent normal tissues. *ZYX* also regulates EMT via the WNK1/SNAI1 signaling to increase the invasion ability of GC cells [44]. A more recent study analyzed the gene expression profiles of GC, intestinal metaplasia, and normal mucosa-non-atrophic gastritis through RNA sequencing. The hub genes of differentially expressed genes between intestinal metaplasia and normal mucosa-non-atrophic gastritis involved *DPP4*, *OLFM4*, *CLCA1*, *SI*, and *MEP1A*. These genes were enriched in protein digestion and absorption and carbohydrate digestion pathways [45]. As described above, the cDNA microarray identified numerous candidate genes associated with invasion. Meanwhile, the candidate genes vary extensively among different studies. Thus, it is essential to isolate the key genes involved in the invasion ability of GC. Advancement of next-generation sequencing may contribute to the identification of the predominant GC genes and pathways. Clinical studies coupled with genotype-matched therapeutic options are required to demonstrate clinical practice.

## 5. Signaling Pathways Involved in GC Invasion and Metastasis

The transfer of extracellular signaling to the nucleus through receptors on the cell surface and cytoplasmic moderators represents a central process for the development of malignancies. Accumulated evidence has demonstrated that various signaling pathways are essential for the invasion of GC. Here we describe the function of the signaling pathway focusing on their roles in regulating cell invasion and metastasis (Figure 3).

### 5.1. RHO/ROCK Signaling Pathway

The small guanosine triphosphatase (GTPase) Rho family such as RhoA, Rac1, and Cdc42 are pivotal modulators of actin assembly, actomyosin contractility, and cytoskeletal rearrangements in the course of malignant cell invasion. ROCK is the downstream effector of the Rho GTPases which regulate cell growth, adhesion, invasion, and apoptosis by modulating cell–cell contacts and cytoskeletal dynamics [46]. The bleb-like protrusions, the construction of actin stress fibers, and facilitated actomyosin contractility are principally mediated by the RhoA and ROCK, whereas Rac1 promotes lamellipodia formation [21]. Rac signals can prompt AMT by the SCAR/WAVE2 conjugate regulation such as actin nucleating, shape, and movement of cancer cells for the plasticity of invasion modes [47].

Different molecules have been demonstrated to modulate the migration mode of cancer cells by regulating the Rho/ROCK signal pathway. Proteinase-activated receptor1 (PAR1) is a member of the G-protein-coupled receptor superfamily. A recent study has implicated that the activation of PAR-1 by thrombin promotes GC cell invasion by the acquisition of morphological change involved in an elongated and polarized morphology, extending pseudopodia at the leading edge by targeting RhoA and Rac1 [48]. RhoJ, one of the members of the Rho GTPases family, was markedly overexpressed in the EMT-subtype, classified by the Asian Cancer Research Group, and closely related to the EMT procedure to enhance the invasion and metastasis of GC cells via IL-6/STAT3 [49]. G17E is a malignant phenotype resulting from RhoA mutations where the frequency is high in GC. Ectopic expression RhoA mutant G17E induced morphological changes, involving the formation of a spindle shape and sharp edges, but did not alter growth in GC cells. G17E upregulated Vav1 expression and facilitated the invasion of GC cells via MMP-9, implying that RhoAG17E/VAV1 signaling could be a potential therapeutic target for GC [50]. Rho GDP differentiation inhibitor 2 (RhoGDI2) can enhance invasion via the Rac1/Pak1/LIMK1 pathway [51]. RhoGDI2 also prompts Rac1 activity mediated by Filamin A, an actin-binding protein, and enhances the binding between Rac1 and Filamin A, which is crucial for the invasion of GC cells [52].

### 5.2. Wnt/β-Catenin Signaling Pathway

The Wnt/β-catenin signaling pathway is an intercellular signaling cascade and has important roles in embryonic progression and adult tissue homeostasis and regeneration [53]. The dysregulation of the Wnt/β-catenin pathway frequently results in a variety of severe diseases, including cancer. An activated Wnt/β-catenin pathway represents an EMT modulatory signaling owing to its regulation of epithelial integrity and tight adherens junctions, and plays a crucial role in enhancing invasion [53]. A recent study has demonstrated that CCT5, a subunit of chaperonin containing TCP1 complex, remarkably enhances GC cell invasion and lymph node metastasis. Mechanically, CCT5 binds to E-cadherin and disrupts the E-cadherin/β-catenin complex, thereby facilitating the nuclear transfer of β-catenin and enhancing Wnt/β-catenin signaling activity and EMT, implying a significant possibility of CCT5 as a biomarker for GC detection [54]. Similarly, Wnt/β-catenin signaling was closely associated with calpain isoform Capn4, which induced cell invasion of GC by promoting MMP9 expression, suggesting that Capn4 could be a possible therapeutic target for GC patients [55]. Upregulation of serum asymmetric dimethylarginine (ADMA), an endogenous nitric oxide synthase inhibitor, enhanced the invasion and EMT, thereby not affecting the growth of GC cells. ADMA prompted the expression of β-catenin and thus activated the Wnt/β-catenin pathway [56]. Notably, a previous study described that Wnt/β-catenin signaling could be regulated by hypoxia-inducible factor (HIF)-1α activity. Wnt/β-catenin pathway regulated urokinase-type plasminogen activator (uPA) and MMP-7 expression, which provided the enhancement of invasion in hypoxic GC cells [57]. Meanwhile, the Wnt/β-catenin signaling is one of the major activated pathways that regulate trastuzumab resistance [58]. Wnt3A, FZD6, and CTNNB1 were upregulated, whereas GSK-3β was downregulated in trastuzumab-resistant GC cell lines compared to their parental cells, suggesting that the acquirement of trastuzumab resistance was closely associated with the activation of Wnt/β-catenin signaling in GC. Interestingly, a blockade of the Wnt/β-catenin signaling by a specific Wnt/&-catenin inhibitor especially reduced the invasion of trastuzumab-resistant cells and reversed EMT [58].

### 5.3. PI3K/AKT/mTOR Signaling Pathway

The phosphatidylinositol-3 kinase (PI3K)/Akt/mammalian target of rapamycin (mTOR) signaling pathways are frequently activated in human malignancies, including GC, and cause carcinogenesis and development [59]. Phosphorylated PI3K/Akt/mTOR signaling promotes many cellular biological activities, such as cell growth, differentiation, intracellular trafficking, and invasion. The tumor suppressor gene phosphatase and tensin homolog (PTEN) plays a critical role in inhibiting tumor proliferation, invasion, and metastasis. A previous study has shown that upregulation of PTEN repressed the PI3K/nuclear factor-kappa B (NF-κB) pathway and restrained the DNA binding of NF-κB on the focal adhesion kinase (FAK) promoter, which led to the inhibition of GC invasion [60]. Bcl2-associated athanogene 4 (BAG4), a member of the human BAG family of proteins, phosphorylated the PI3K/AKT/NF-κB/ZEB1 axis and enhanced the invasion and metastasis of GC cells. The function of the PI3K/AKT signaling pathway is significantly correlated with the BAG4-induced EMT and the expression of ZEB1 [61]. Meanwhile, silencing *ROS1*, an oncogene that encodes a tyrosine kinase of the insulin receptor family, attenuated GC cell invasion and EMT through the PI3K/Akt signaling pathway [62]. Carcinoembryonic antigen-related cell adhesion molecule 6 (CEACAM6), a glycosylphosphatidylinositol-linked immunoglobulin superfamily member, has pivotal oncogenic roles by inducing EMT as validated by upregulations in the EMT markers N-cadherin, Vimentin, and Slug. CEACAM6 also promotes the production of MMP-9 through the PI3K/AKT pathway in GC cells [63]. Similarly, casein kinase II (CSNK2), a pleiotropic serine/threonine kinase, acts as an oncogene in GC invasion via EMT and the PI3K/Akt/mTOR signaling pathway [64]. Salvianolic acid B (Sal-B), a chemopreventive agent that suppresses oxidative stress and apoptosis, downregulated EMT to restore the resistance to cisplatin by promoting AKT/mTOR pathway in cisplatin-resistant GC cells [65].

### 5.4. JAK/STAT Signaling Pathway

Cytokines including interleukin (IL), interferon (INF), chemokines, and tumor necrosis factor connect to subsequent receptors to stimulate signaling pathways, including progressive preserved signaling namely, the Janus Kinases (JAK)/Signal Transducer and Activator of Transcription (STAT) signaling pathway. The JAK-STAT pathway controls a multitude of biological functions, such as immune-system development, stem cell continuity, and tumor initiation [66]. Accumulating evidence has reported that JAK/STAT signaling had a significant role in the invasion of GC [67]. Previous reports described that the upregulation of gastrokine2, a gastric tissue-specific tumor suppressor, inhibited the invasion of GC cells by suppressing the JAK2/STAT3 signaling pathway, and downregulated MMP-2 and MMP-9 activity [68]. Growth differentiation factor 15, a member of the TGF-β superfamily, promoted invasion and EMT via STAT3 activation in refractory GC cells [69]. Interestingly, leptin, an adipocyte-derived hormone, activates the invasion of GC cells by targeting the MEK as well as the JAK-STAT pathways, which led to the preservation of stemness and metastatic capability, indicating that leptin-mediated signaling could serve as a potent therapeutic target for GC [70].

### 5.5. NF-κB Signaling Pathway

The nuclear factor-κ-gene binding NF-κB pathway, a family of transcription factors, has critical roles in various pathological processes of malignancies, including cancer stemness, drug resistance, and metastasis [71]. Different signaling pathways, which are induced by various growth factors, tyrosine kinases, and cytokines, are responsible for NF-κB activation. Emerging evidence has demonstrated that NF-κB signaling can act as an upstream regulator to prompt the EMT phenotype for enhancing the invasion of GC. ADAMTS16, a member of a disintegrin and metalloproteinase with thrombospondin motifs protein family, enhanced invasion by targeting IFI27 via the NF-κB pathway [72]. ADAMTS16 upregulated NF-κB associated proteins, such as IκBα, and P65. By contrast, ADAMTS19 restored cell migration and invasion by targeting a member of the EF-hand Ca2 +-binding proteins, S100 calcium-binding protein A16 (S100A16), through the NF-κB pathway [73]. S100A16 also served as a target gene modulating EMT. Alpha B-crystallin (CRYABZ) is a member of the small heat shock protein family. In GC, cell invasion was promoted by overexpression of CRYABZ via the NF-κB-regulated EMT [74]. Likewise, the carboxyl terminus of Hsc-70-interacting protein (CHIP), as a U-box-type ubiquitin ligase, acts as a tumor suppressor. Upregulation of CHIP markedly restored the invasion of GC cells through NF-κB subunits, RelA/p65, and RelB signaling by reducing TRAF2 activity. The impaired TRAF2-NFκB axis suppressed Bcl-2, MMPs, and β-integrin expression, which may be closely related to the reduced invasion of GC cells [75]. Hexokinase domain component 1 (HKDC1) has recently been identified and categorized as a fifth hexokinase and is critical to preserving whole-body glucose homeostasis. HKDC1 also acts as an oncogene which has a pivotal role in stimulating EMT of GC cells by activating the NF-κB pathway. Additionally, HKDC1-regulated chemosensitivity of GC cells to cisplatin, oxaliplatin, and 5-fluorouracil (5-FU) was responsible for DNA damage repair, which led to further activation of NF-κB signaling [76]. A gram-negative, spiral-shaped bacterium *Helicobacter pylori (H.pylori)* infection has been said to be the highest risk factor of GC in clinical–epidemiological studies [77]. *H. pylori* is identified for more than 60% of GCs. To date, *H. pylori* can be detected by a rapid urease test, histological findings of biopsy samples, and the polymerase chain reaction technique [78]. Recent studies have demonstrated that *H. pylori* infection is not only an important factor in GC carcinogenesis but is also closely associated with the invasion and metastasis of GC [79,80]. Zinc finger and BTB domain containing 20 (ZBTB20), a sequence-specific transcriptional suppressor, was significantly overexpressed in GC cells under the existence of *H. pylori*. ZBTB20 also promoted invasion and MMP-2/-9, and NF-κBp65 activation, whereas ZBTB20 attenuated the IκBα expression of GC cells [81].

### 5.6. Transforming Growth Factor-β Signaling Pathway

TGF-β is a principal effector in GC and regulates the transition of fibroblasts to CAFs, enhancing cell invasion via the promotion of EMT [82,83]. A recent study described that transient receptor potential vanilloid 2 (TRPV2), a highly Ca^2+^-permeable ion channel, enhanced GC cell invasion by upregulation of RUNX2, TGFBR1, and JUN, which were closely related to TGF-β signaling [84]. Stimulation of recombinant TGF-β led to an evident increase in a specific metastasis-related protein and S100A4 expression level, and enhanced the invasion ability of GC cells, which resulted in activation of the Smad signaling pathway, thereby increasing epithelial markers and reducing mesenchymal markers [85]. Similarly, inhibin βA, a protein-coding gene and member of the TGF-β superfamily, enhanced the invasion of GC by activating the TGF-β signaling pathway [86]. Nicotinamide N-methyltransferase (NNMT) is a metabolizing enzyme that mainly catalyzes the methylation of nicotinamide (vitamin B3) and other pyridines into pyridinium ions [87]. Recent studies demonstrated that NNMT had a pivotal role in GC carcinogenesis and acted as a promising diagnostic and prognostic biomarker [87,88]. Furthermore, several NNMT inhibitors are already available and could be tested for GC management [89,90,91]. The upregulation of NNMT expression in GC cells led to a significant increase in the level of EMT markers and the promotion of invasion. Interestingly, the alteration of NNMT expression was closely associated with TGF-β1 activity, thus implying that EMT promotion is regulated by the induction of NNMT [92]. By contrast, *Dachshund homolog 1 (DACH1)*, a homolog of Drosophila dac in humans, impaired GC invasion and EMT by suppressing TGF-β signaling [93]. Elucidating the mechanism of how TGF-β induces the invasion capacity of GC cells through EMT may help to find potent targets for therapy for GC.

### 5.7. RAS/RAF/ERK/MAPK Signaling Pathway

The RAS/RAF/ERK/MAPK signaling pathway is one of the most familiar signalings in the pathology of cancer. *RAS* and *BRAF* are members of this pathway and can regulate cellular reactions to growth signaling. MAPK is one of the critical signaling pathways involved in the regulation of growth, invasion, angiogenesis, and metastasis. Furthermore, RAS/RAF/ERK/MAPK signaling can interfere with several other signalings, including PI3K/AKT/mTOR and NF-κB. SIRT2 is a member of the sirtuin homologs, which function as NAD+-dependent class III histone deacetylases. RAS and its downstream ERK/JNK/MMP-9 have a critical role in the SIRT induced invasion ability of GC cells by prompting phosphoenolpyruvate carboxykinase 1-related metabolism [94]. TATA-box-binding protein-associated Factor 15 (TAF15), a member of the FUS/EWS/TAF15 (FET) family, promotes the migration and invasion of GC cells through RAF1/MEK/ERK signaling [95]. Membrane-associated guanylate kinase inverted 1 (MAGI1) acts as a tumor suppressor, and silencing MAGI1 notably facilitated the invasion of GC cells by promoting the expression of EMT-related molecules and MMPs by impairing MAPK/ERK signaling [96]. In addition, the activity of MMP-1. -2, and -9, stimulated by cytokines including TNF and IL-1β, are prompted by the MAPK ERK1/2 signaling pathway [97,98,99]. Transduction of oncogenic KRAS^G12V^ upregulated spheroid formation ability as well as CD44 and Sox2 expression, leading to the acquisition of cancer stem cell (CSC) phenotypes in wild-type KRAS-expressed GC cells. RTK-RAS signaling promotes EMT in GC cells, leading to the acquisition of CSC phenotypes and metastatic capacity, and resistance to 5-fluorouracil and cisplatin [100].

### 5.8. Another Signaling Pathway

The Hippo pathway has been shown to have a pivotal role in the development and homeostasis of tissue organs and positively regulates the differentiation, growth, apoptosis, and invasion of cancer cells [101]. This pathway involves two main downstream substrates, Yes-associated protein1 (YAP1) and transcriptional co-activator with a PDZ-binding motif. The disruption of Hippo signaling is a frequent event and has been reported to be closely associated with the invasion of GC. A recent study demonstrated that extensive expression of GNB4, one of the essential elements of heterotrimeric G proteins, induced by *H. pylori* infection prompted the invasion of GC by stimulating the YAP1 expression and its target genes, CYR61 and CTGF, indicating that GNB4 prompted the Hippo-YAP1 pathway in GC cells [102]. F-box and WD repeat domain-containing 5 (FBXW5), the E3 ubiquitin ligase, downregulated the activity of the Hippo signaling pathway by promoting large tumor suppressor kinases 1 ubiquitination, and degradation, leading to indirect YAP activation and thereby increased invasion, EMT, and metastasis of GC cells [103].

HIF-1α has an important role in the activity of GC induced by hypoxia [104]. GC cells treated with hypoxia revealed enhanced invasion capacity and reduced expression of E-cadherin and N-myc downstream-regulated gene 2 by stimulating HIF-1α and Twist [105].

## 6. Tumor Microenvironment and GC Invasion

Over the last several decades, it has become distinct that cancer invasiveness is defined not only by cancer cells but also by the tumor microenvironment (TME) [106]. It comprises a complicated network of cancerous and non-cancerous cells that prompt invasion and metastasis and regulate the efficacy of therapy for cancer [106]. Particularly, the TME consists of various components, including cancer cells, ECM, BM, tumor-associated macrophages (TAMs), cancer-associated fibroblasts (CAFs), immune cells (T cells, B cells, neuroendocrine cells, dendritic cells (DCs), myeloid-derived suppressor cells (MDSCs)), and blood and lymphatic vessels [107]. Cancer cells produce a variety of growth factors and cytokines, and inflammatory and matrix-remodeling enzymes [108]. The TME also has a pivotal role in carcinogenesis and pre-metastatic niche formation, leading to the enhancement of the capacity of cancer cells to invade and disseminate. The ECM is the chief structural element of the TME. For cancer cells, the capacity of malignant cells to invade needs the degradation of the ECM which is involved in ECM remodeling in collective and mesenchymal cell invasion mode.

Proteolytic activity is the first process in degrading the BM which leads to invasion of adjacent tissues. This process is accomplished via the production of specific proteases including MMPs, uPA, and other specific-targeting proteases. These proteases are extensively overexpressed in GC and are often related to poor survival. For instance, in early GC the expression of MMP-1 mRNA was upregulated, particularly in *H. pylori*–infected patients [109]. MMP-1 expression in GC cells at the leading edge of invasive tumors can predict lymph node metastasis [110]. By contrast, MMP-1 may have an anti-oncogenic role in GC progression, although it facilitated proteolytic activity in vitro [111]. The expression of MMP2 was correlated with invasion, metastasis, and microvessel density in GC [112]. Furthermore, MMP2 can facilitate angiogenesis in GC [113]. A recent study described that the adenosine triphosphate synthase F1 β subunit in the TME promotes the invasion activity of GC cells through the FAK/AKT/MMP2 pathway [114]. Upregulation of MMP9 has been shown in GC and its expression is closely correlated with clinicopathological findings [115]. The level of MMP9 in serum enhanced progressively depending on the depth of tissue invasion in patients with GC [116]. A recent study presented that downregulation of NDRG1, 43 kDa protein, which is ubiquitously expressed in human tissues, enhanced the invasion of GC cells by targeting MMP-9 [117]. MMP-7, shown to be overexpressed in the epithelium in patients with *H. pylori-*infected gastritis, positively correlated with gastric tumorigenesis [118]. MMP-7 upregulation is associated with tumor size, invasion, and microvessel density, which lead to the formation of metastasis [119]. Upregulation of MMP-7 results in enhanced levels of activator protein 1, gastrin, which is closely associated with *H. pylori* infection [120]. MT-1MMP, known as MMP-14, functions as one of the crucial drivers of cancer invasion and metastasis. MMP-14 was shown to promote cancer progression by stimulation of proMMP-2 and degradation of ECM [121]. The expression of MT1-MMP was increased in GC cells, and MT1-MMP promoted the invasion of cells by modulating the vimentin and E-cadherin expression [122].

The urokinase-type plasminogen activator receptor (uPAR) is the cell surface receptor for the extracellular serine protease uPA. uPAR modulates the proteolytic activity of the ECM by connecting uPA to activate the plasminogen activation system as well as introducing intracellular signaling pathways including cell growth, adhesion, and apoptosis with other molecules, such as integrins [123]. An earlier study demonstrated that upregulation of uPAR was especially shown in poorly differentiated GC [124]. The upregulation of semaphorin 5A, which is a member of class 5 of the semaphorin family, promoted GC cell invasion by increasing uPA expression through the PI3K/Akt signaling pathway [125].

CAFs, the stimulated fibroblasts in surrounding cancer stroma, are considered to comprise a dominant stromal component of cancer cell invasion by remodeling the ECM [126]. CAFs have been shown to promote cancer invasion by inducing intimate interaction with cancer cells and modulating cancer-related signaling [127]. Activated CAFs can secrete a fair lot of soluble factors, involving chemokines, cytokines such as CXCL12 (SDF1), CXCL14, IL-6, IL-23, IL-33, and ligands of epidermal growth factor receptor (EGFR), members of the vascular endothelial growth factor (VEGF) family, basic fibroblast growth factor (bFGF), platelet-derived growth factor (PDGF), TGF-β, and insulin-like growth factor (IGF) [128,129]. These soluble factors can increase tumor invasion [130]. A recent study has shown that CXCL5 stimulates GC cells to induce EMT, therefore regulating pre-cancerous activation of neutrophils, which facilitates the invasion ability of GC cells [131]. The CXCL10/CXCR3 axis enhances GC cell invasion by stimulating MMP-2/9 secretion through the PI3K/AKT pathway, suggesting that a CXCR3 could be a candidate for targeted therapy for GC patients [132]. IL-33, a member of the IL-1 superfamily, facilitates the invasion and EMT of GC cells triggered by CAFs by the ERK1/2-SP1-ZEB2 signaling activity through ST2L [133]. Similarly, the invasion of GC cells was enhanced by IL-23 connecting to its receptor and thereby induced microtubules via the STAT3 signaling [134]. Fibroblast activation protein (FAP), a type II integral membrane gelatinase of the serine protease family, is one of the CAFs markers which exists in the stromal element. Stromal FAP secreted from CAFs facilitates GC cell invasion through EMT phenotype which is closely correlated with the Wnt/β-catenin pathway, because the expression of DKK1 and LEF-1 protein, which are involved in Wnt/β-catenin signaling, were enhanced with exogenous FAP [135].

CAFs also enhance the invasion of cancer cells through the BM and make gaps in stromal elements and the BM in an MMP-independent manner [136]. Additional study shows that CXCL12 derived from CAFs enhances the invasion of GC cells via stimulating the β1-integrin clustering at the tumor cell surface [127]. Additionally, CAF-mediated TME remodeling promotes the invasion of cancer cells. EMT is facilitated by the collagen-rich matrix, which leads to promoting the invasion of GC cells [25]. Remarkably, CAFs may produce gaps in the BM and stromal components which are bound to cell–cell junctions to prompt collective cell invasion in an MMP-dependent or MMP-independent manner [136,137]. Molecules derived by CAF in the TME could be utilized as a therapeutic approach for GC, which yields novel insight for the advancement of GC treatment.

## 7. The Role of Non-Coding RNA Invasion of GC

Most RNA transcribed but not encoded proteins can be characterized as non-coding RNAs (ncRNAs) [138]. Depending on the shape, length, and location, ncRNAs have been subdivided into distinct classes, such as microRNA (miRNA), long ncRNA (lncRNA), circular RNA (circRNA), and PIWI interacting RNA (piRNA). They can mediate transcription factors to bind to promoters and thereby modulate the advancement of various malignancies [138]. ncRNAs have served as important mediators in GC progression by regulating the expression of their target genes. Usually, ncRNAs are shown to be up-or downregulated in cancer specimens compared to normal tissues. Accumulating evidence has implicated that the invasion ability of GC was regulated at the transcriptional level utilizing ncRNAs (Figure 3). ncRNAs are used as a therapeutic target as well as a biomarker for early detection of GC [139,140]. The regulatory processes of ncRNAs are distinct. ncRNAs act as both invasion promoters and inhibitors.

### 7.1. ncRNAs Promoting GC Invasion

A previous study reported that miR-93 promoted cell invasion and stimulated EMT in GC cells. The levels of TIMP2 mRNA and protein were significantly upregulated in the miR-93 inhibitor group, suggesting that TIMP2 was a direct target of miR-93 [141]. MiR-192-5p facilitated EMT and invasion and FOXP3^+^ regulatory T cell differentiation of GC by targeting the RB1/NF-κBp65/IL-10 pathway [142]. Tumor suppressor Programmed Cell Death 4 (PDCD4), which is related to disease progression and poor survival of malignancies, is often downregulated in GC. Attenuated PDCD4 and PTEN expressions were induced by miR-21, which prompted the invasion ability of GC cells [143]. Similarly, tumor suppressor PTEN was downregulated by miR-214 containing the PTEN-3′-UTR construct at the post-transcriptional level, which led to enhanced invasion of GC cells [144].

A recent study explored the activation of the lncRNA LINC00511 on the invasion and progression of GC. Silencing LINC00511 repressed tumorigenesis and invasion of GC cells by sponging miR-515-5p [145]. An easer enzyme, alkylation repair homolog protein 5 facilitates invasion and metastasis by reducing methylation of the long noncoding RNA (lncRNA) nuclear paraspeckle assembly transcript 1 (NEAT1) [146]. Neural invasion represents the proliferation of cancer cells in surrounding nerves and is correlated with the shortened outcome in GC [147]. LncRNA DIAPH2-AS1 was highly expressed in neural invasion-positive GC specimens and facilitated the invasion and NI potential of GC cells by regulating the expression of NTN1 via attachment with NSUN2 [148]. LINC00473 serves as an oncogenic lncRNA to facilitate the invasion of GC cells by targeting MMP2 and MMP9 [149].

CircNRIP1 is highly expressed in human GC specimens and contributes to the invasion of GC cells. CircNRIP1-miR-149-5p-AKT1/mTOR axis is related to the changed metabolism, and increases invasion in GC cells [150]. CircPDIA4 interacted with DHX9 which resulted in the suppression of its repressive activities on circRNA biogenesis in the nucleus and prompted GC invasion. Meanwhile, Cytoplasmic circPDIA4 prevented DUSP6-modulated ERK1/2 downregulation via attaching to ERK1/2 by targeting the MAPK signaling pathway. Interestingly, the efficacy of GC cells on ERK inhibitors was facilitated by silencing circPDIA4 [151].

CAF-derived exosomes may be a positive modulator of scirrhous GC, which was surrounded by abundant fibrotic stroma. CD9 is a specific marker of exosomes that are produced from CAFs, and CD9-expressing exosomes from CAFs increase the invasion ability of GC cells via the activity of MMP2 [152]. Exosomal circ-RANGAP1 expression was enhanced in patients with GC. Upregulation of Circ-RANGAP1 facilitated GC cell invasion via the miR-877-3p/VEGFA axis [153].

### 7.2. ncRNAs Inhibiting GC Invasion

MiR-3648 negatively contributes to the invasion ability of GC via targeting FRAT1 and FRAT2, a negative modulator of the Wnt/β-catenin signaling. Upregulation of miR-3648 was negatively related to c-Myc, FRAT1, and FRAT2 expression, indicating that miR-3648 serve as a tumor-suppressive miRNA [154]. MiR-200b-3p suppressed the invasion and lung metastasis of GC cells and downregulated the expression of CXCL12 and CXCR7 [155]. Upregulation of miR-214 in CAFs has revealed enhanced E-cadherin expression and repressed Vimentin, N-cadherin, and Snail expression. Also, overexpression of miR-214 caused the inhibition of invasion and metastasis via targeting FGF9 in GC cells [156]. It was reported that Circ-CEP85L upregulation in the GC cell line resulted in the inhibition of invasion and migration by modulating NFKBIA which was a direct target of miR-942-5p [157]. Exosomal miR-139, downregulated in CAFs, facilitated the MMP11 expression, leading to increased invasion and metastasis of GC cells [158].

ncRNAs contributing to invasion and metastasis may serve as a useful tool for treating GC invasion and metastasis. Table 1 shows an overview of ncRNAs that prompt and inhibit the invasion and metastasis of GC for the past five years.

## 8. Therapeutic Implications for Targeting the Invasion of GC

Acquiring knowledge of mechanisms regulating invasion has identified several conceivable applications for the management of GC. The most basic strategy is to focus on targeted therapies specific to invasion. The increased expression of proteases which degrade ECM is considered one of the most evident and long-noted changes in cancer invasion by the mesenchymal process after EMT, and is thereby identified as a target for therapy. Andecaliximab (GS-5745), a monoclonal antibody targeting MMP-9 activity, has been evaluated in a phase Ib trial in combination with mFOLFOX6. Significant improvement in progression-free survival (PFS) and objective response rate (ORR) was observed in patients with gastric or gastroesophageal junction adenocarcinoma [183]. Nevertheless, overall survival (OS) was not improved in a phase III clinical study of these malignancies [184]. In this way, abundant MMP inhibitors were evaluated in clinical studies for GC, but none of these approaches provided the expected results. The reasons for failure may be broad spectrum inhibition of MMPs and deficiency of acknowledgment of MMP selection which would be targeted, undesirable side effects, poor pharmacokinetic properties, poor knowledge of MMP biology, and animal models not paralleling human disease [185]. Inhibition of the useful roles of MMPs caused undesirable advertising events.

Meanwhile, ATF-Fc, formed by linking ATF and the human IgG1 Fc fragment, suppresses the metastasis of GC cells by abolishing the relation of uPA/uPAR and impairing tumor neo-vascularity [186]. Furthermore, combined ATF-Fc with trastuzumab further blocks the metastasis of HER-2-positive breast cancer cells by destroying the interference with the uPA/uPAR and HER-2 signaling [187]. A more recent preclinical study has shown that a combined therapy of anti-uPAR and anti-PD1 significantly reduces the growth of tumors and improves survival in diffuse-type GC via the ERK pathway [188]. Furthermore, uPAR CAR-T cells in combination with PD-1 inhibition more effectively reduced GC growth and prolonged survival in both the cell line-derived xenograft and patient-derived xenograft models. Nevertheless, these uPA/uPAR inhibitors have yet to further establish their clinical efficacy.

The Rho/ROCK pathway has a key role in cytoskeletal rearrangement in almost all modes of invasion, and thereby seems to be a useful target for therapy. A previous study showed that miR-31 repressed the invasion of GC cells by directly targeting RhoA, indicating that miR-31 could be a latent therapeutic target for GC metastasis [189]. Lately, a series of novel RhoA inhibitors to improve their biological feature has been created [190]. Among them, JK-206 markedly suppresses viability as well as migration in GC cells. Furthermore, administration of JK-206 mediated pathways involves the Myc target, G2/M checkpoint, and microtubule dynamics [191]. Meanwhile, miR-381 suppressed the invasion ability of GC cells by targeting ROCK2 and downregulating MMP-2 and MMP-9 [192]. Likewise, miR-148a inhibits GC cell invasion activity by targeting ROCK1 [193]. Notably, AGC kinases are a congregate of serine–threonine kinases which comprise PDK1, PKA/PKB, and S6K. AT13148, an inhibitor of AGC kinase with dominant ROCK and AKT suppression, was evaluated in a global phase I clinical study in patients with advanced solid tumors. Unfortunately, significant clinical responses were not shown in this trial and their clinical use could be restricted by advertising events related to inhibition of ROCK [194].

Although current evidence shows the feasibility of the use of the Rho/ROCK pathway as a druggable target, more study of its application for the therapy of GC is immediately required. Another Rho GTPases family member, Rac may also be an attractive target specific to invasion, due to its central role in cancer aggressiveness through the mediation of cytoskeletal dynamics, invasion, metastasis, EMT, and cell–ECM interactions. Rac1 inhibitor NSC23766 suppresses the invasion, EMT, and cancer stem-like cell phenotype, thereby inhibiting metastasis and resistance to chemotherapy [195].

As mentioned above, although numerous studies targeting ECM-degrading proteases have been evaluated, almost all of them were unsuccessful in improving the OS of patients with GC [196]. These disappointing results may be due to the capacity of the cancer cells to switch between diverse invasion modes during the metastatic process. Invasion plasticity is a pivotal feature of cancer cells that makes them able to resist management that targets a specific invasion mode, and thus strategies designed to inhibit plasticity by blocking various drivers that induce resistance may result in successful treatments [197]. To resolve these challenges, therapies targeting cytoskeletal modulators that are involved in various modes of invasion (or a combination targeting different main drivers) are required. One attractive candidate for key regulators may be β-integrin, which regulates various invasion modes. A recent study demonstrated that β1-integrin upregulates the infiltration that is involved in the abnormal cell–ECM interactions in E-cadherin dysfunctional GC cells [198]. In addition, several therapeutic medications are being evaluated in patients with GC. For instance, it was recently observed that the αvβ5-integrin inhibitor SB273005 repressed the formation of filopodium and cell migration that was induced by one of the members of the Rho GEF family, pleckstrin domain protein 1 (FARP1) [199]. It was also demonstrated that miR-124-3p acts as a tumor suppressor and suppresses the invasion of GC cells by targeting β3-integrin, and this was validated by a bioinformatic analysis [200]. There have been no clinical trials of integrin-based therapies for GC, which may be due to the limited data regarding the integrin expression profiles of patients with GC.

Cdc42 and its effecters may also be candidates because they are involved in both elongated-mesenchymal and rounded-amoeboid invasion strategies. For example, miR-133 restored the expression of Cdc42 by specifically and directly connecting to its 3′UTR and thereby inhibiting the Cdc42/PAK pathway, which impeded the invasion ability of GC cells [201]. Another preclinical study showed that miR-497 directly targeted CDC42 and modulated the migration and invasion of GC cells via β1-integrin /FAK/PXN/AKT signaling [202]. These studies imply that (i) ncRNAs may be the critical regulators of CDC42, and (ii) the tumor-suppressive properties of ncRNAs may be an attractive approach for preventing the initiation and progression of GC.

ECM remodeling may also contribute to GC treatment along with integrin-targeted approaches. Several types of cancer have exhibited an unusual deposition or accumulation of ECM elements and promoted ECM stiffness, which reduces drug infiltration and effectiveness for therapy [203]. In addition, an enhanced ECM density promotes cell–ECM binding and thus increases the growth of GC cells [204]. Targeting ECM-associated enzymes including collagenase, MMPs, or lysyl oxidases may therefore be a useful therapeutic approach for GC. As mentioned above, MMP inhibitors produced unsuccessful results in all trials in which they were examined, but specific MMP-targeting strategies utilizing monoclonal antibodies have been described, and their efficiency and tolerance could be developed [205]. Concerning therapeutic strategies that directly aim at ECM elements, the blockade of fibronectin (FN) biosynthesis is one of the most promising candidates. pUR4B, a small peptide inhibitor, efficiently impairs the polymerization of FN in the ECM [206]. In co-cultures of breast cancer cells and fibroblasts, pUR4B reduced the deposition of FN in the ECM and other stromal proteins [207]. The assembly of the actin cytoskeleton is the major cellular process that regulates the invasion of cancer cells [208], and it was recently reported that in a specific condition, the altered expression of actin cytoskeletal regulators (i.e., Arp2/3, ADP/cofilin, fascin, filamin A, actinin, and tropomyosins) or non-actin cytoskeletal regulators (i.e., FAK, vimentin) can facilitate the MAT [209]. It thus seems possible to use small molecules targeting either actin or its dynamics as a therapeutic approach. However, at present, no therapeutic strategy has been validated in examinations of GC.

## 9. Discussion and Future Directions

Because of its most defining feature of malignancy, the invasion characteristics of GC are expected to be located at the key process of cancer metastasis and can be considered a possibly exclusive therapeutic target. In this review, we summarized the various factors affecting the invasion of GC cells, such as proteases, signaling pathways, genetic alteration, TME, and ncRNAs (Table 1 and Table 2).

Targeting invasion plasticity has been investigated over the last 10 years [197]. In ovarian cancer cells, dual cisplatin/paclitaxel-resistant cells were shown to invade through both mesenchymal and amoeboid phenotypes which were mediated by the Rho/ROCK pathway. An upregulation in the phosphorylated myosin light chain was detected in drug-resistant cells, suggesting enhanced actomyosin contractility. Notably, the knockdown of both two non-muscle isoforms of non-muscle myosin II (NMM IIA and IIB) led to an inhibition of protease activity, with near complete abolition in dual knockdown cells. These results suggest the prospect of inhibiting the invasion of cancer cells by targeting NMM II. GC treated with progressive doses of vincristine were observed to switch to amoeboid transition, and targeting NMM II may therefore provide a prospective future direction for GC therapy. The scaffolding protein NEDD9 is required for both mesenchymal and amoeboid invasion in triple-negative breast cancer [211]. NEDD9 deficient breast cancer cells demonstrate amoeboid morphology and a robust decrease in active Rac1. The dual targeting of two distinct invasion modes through NEDD9 and ROCK/RhoA contributed to further inhibition of the invasion/metastasis of breast cancer cells [211]. WNT5A is a glycosylated and lipid-modified secreted protein ligand that triggers mainly the activation of noncanonical WNT signaling pathways. A recent study explored how WNT5A activity relates to the invasion mode of melanoma cells [212]. Suppressed WNT5A signaling led to a MAT that reduced the Cdc42-modulated invasion. Remarkably, dual targeting of WNT5A and RhoA activity revealed a more efficient approach to inhibiting melanoma cell metastasis compared to the restoration of individual targets alone [212]. Increasing evidence has indicated that the biological properties of microtubules, an element of the cytoskeleton, and their dynamics are closely correlated with invasion plasticity [213]. Microtubule drugs hamper the invasion plasticity that regulates diverse efficiencies in each invasion mode. Drugs blocking microtubule assembly induce the amoeboid invasion and subsequent RhoA activation. These findings imply that a combination of microtubule drugs, such as vincristine, and specific invasion inhibitors could synergistically target both the amoeboid and mesenchymal invasion [213].

These combination strategies will open a new avenue for achieving a synergistic blockade of invasion plasticity in the treatment of GC. However, further studies evaluating the mechanisms that underlie invasion plasticity are urgently needed.

## 10. Conclusions

In summary, this review has systematically illustrated molecular insight related to the mechanisms responsible for GC tumor invasion and metastasis. Metastasis is the primary cause of cancer-related deaths from GC. Thus, it is important to study the mechanisms that underly tumor invasion plasticity to validate novel therapeutic targets. This review also explored and summarized the signaling pathways that regulate GC cell invasion and metastasis. Accumulated evidence reveals that gene alterations might constitute especially interesting targets for combatting the invasion of GC. The invasion of GC cells and their ability to develop metastases are modulated by mechanisms driven by the TME. Specifically, CAFs can produce various factors to prompt GC invasion and metastasis, involving chemokines and cytokines, and to remodel the ECM. Recent studies have also demonstrated the critical role of ncRNAs in promoting tumor invasion in GC. Many ncRNAs have been revealed as underlying useful tools for treating the invasion and metastasis of GC. Compared to conventional therapy using protease or molecular inhibitors alone, multi-therapy targeting invasion plasticity may seem to be an assuring direction for the progression of novel strategies in the future. More comprehensive analyses of the mechanisms underlying the regulation of invasion plasticity will be crucial to identifying innovative therapies and potent curative approaches for GC.

## Figures and Tables

**Figure 1 cancers-16-00054-f001:**
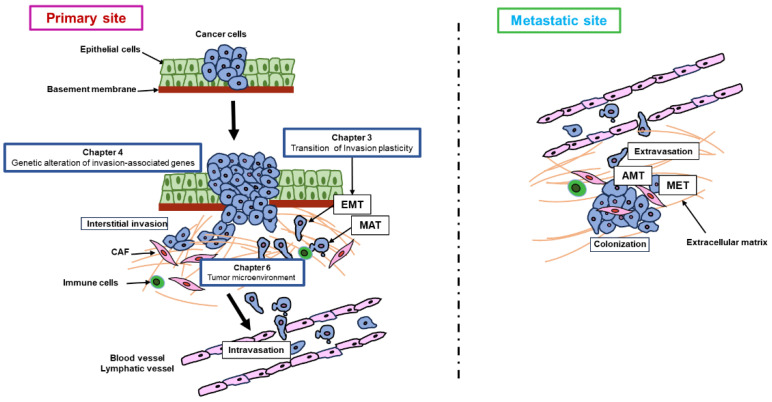
Schematic diagram of gastric cancer invasion and metastasis. Cancer cells occur within the gastric epithelium, destroy the basement membrane, and then enter the submucosa due to a repression of intercellular adhesion molecules and a high motile ability. Cancer cells infiltrate the surrounding tissues. The complexities of the metastatic process consist of several stages: transition in blood or lymph, invasion into the systemic circulation, extravasation, and colonization in the metastatic site. The reasons restraining the proliferation of malignant cells involve the BM and various elements of the surrounding stroma, and consistent revelation to immune cells. Chapter numbers shown in this figure represent how these chapters relate to the process of invasion and metastasis.

**Figure 2 cancers-16-00054-f002:**
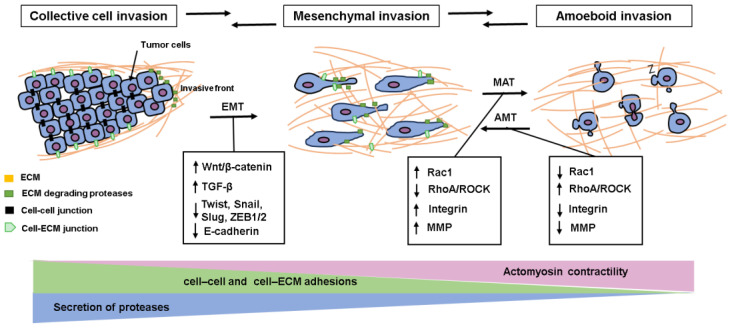
Schema of invasion plasticity. In collective cell invasion, cells can maintain cell–cell attachments and invade collectively, harmonized as sheets or cell clusters. Epithelial–mesenchymal transformation (EMT) is a molecular mechanism characterized by the loss of polarity of epithelial cells, which disrupts cell attachment to the extracellular matrix (ECM). EMT leads to increased invasion capacity and histologically acquires a more mesenchymal phenotype, which represents an elongated, spindle-like cell shape with the formation of pseudopod protrusions and filopodia. Alternatively, by losing dependence on ECM and by promoting actomyosin contractility, mesenchymal cells can undergo a mesenchymal–amoeboid transition (MAT) and invade in the amoeboid mode. MAT in malignant cells can be induced by decreased cell–ECM interaction, loss of ECM proteolysis, enhanced contractility, the blockade of Rac activity, or indirectly promoting Rho/ROCK activation. The amoeboid and mesenchymal modes of invasion are often inter-convertible, and amoeboid cells can also revert to mesenchymal mode by amoeboid–mesenchymal transition (AMT). Important mediators of these transition changes in expression are described.

**Figure 3 cancers-16-00054-f003:**
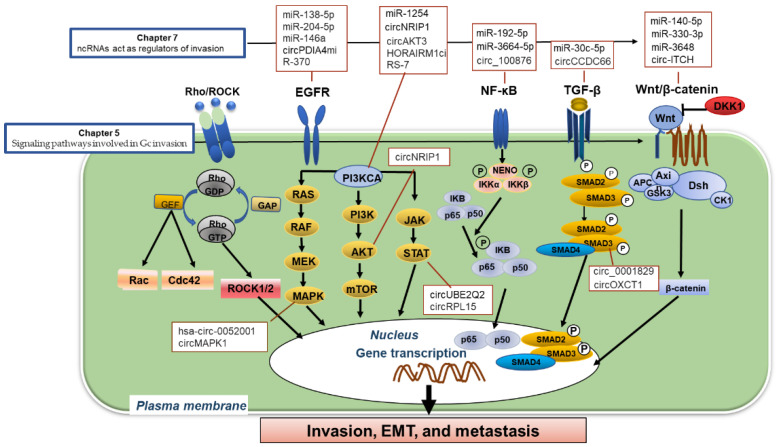
Non-coding (nc)RNA-regulated signaling pathways involved in gastric cancer invasion. The function of the signaling pathway focusing on their roles in regulating cell invasion is summarized. The experimentally identified ncRNAs act as regulators of several principal pathways involved in invasion–metastasis programs. Chapter numbers shown in this figure represent how these chapters relate to the process of invasion and metastasis.

**Table 1 cancers-16-00054-t001:** ncRNAs associated with the invasion ability of GC for the past five years. ncRNAs = non-coding RNAs, EMT = epithelial–mesenchymal transition, Treg = regulatory T cell, GC = gastric cancer, MTA1 = metastasis-associated protein 1, SKA21 =spindle and kinetochore associated 2, lncRNA = long non-coding RNA, ALKBH5 = alkylation repair homolog protein 5, NEAT1 = nuclear paraspeckle assembly transcript 1, PCGEM1 = prostate cancer gene expression marker 1, SNHG8 = nucleolar RNA host gene 8, SNHG7 = small nucleolar RNA host gene 7, circRNA = circular RNA, STAT3 = signal transducer and activator of transcription 3, ↓ = downregulation; ↑ = upregulation, N.A. = not applicable.

ncRNAs	Dysregulations	Target	Comments	Ref
**ncRNA prompting invasion**				
**miRNA**				
miR-21	↑	PDCD4, PTEN	Stimulates invasion by attenuating PDCD4 and PTEN expression.	[143]
miR-30c-5p	↓	MTA1	Activates M2 macrophages and, in turn, EMT through TGF-β/Smad2 signaling.	[159]
miR-93	↑	TIMP2	Prompts invasion and EMT by targeting TIMP2,	[141]
miR-192-5p	↑	RB1	Induces EMT and Treg cell differentiation by modulating IL-10 secretion.	[142]
miR-214	↑	PTEN	Post-transcriptionally downregulates PTEN expression and prompts invasion.	[144]
**lnc RNAs**				
NEAT1	↑	ALKBH5	ALKBH5 prompts invasion and metastasis by reducing methylation of the NEAT1.	[146]
PCGEM1	↑	SNAI1	Hypoxia-cultured GC-derived PCGEM enhances invasion.	[160]
DIAPH2-AS1	↑	NSUN2 NTN1	Enhances the neural invasion via the DIAPH2-AS1-NSUN2-NTN1 axis.	[148]
NSLT-1	↑	NOTCH1	Enhances invasion and metastasis by regulating NOTCH1.	[160]
LINC00511	↑	N.A.	Promotes invasion by modulating miR-515-5p.	[145]
SNHG8	↑	PDGFRA.	Enhances invasion by regulating the miR-491/PDGFRA.	[161]
SNHG7	↑	Snail	Increases invasion via repressing the miR-34a-Snail-EMT axis.	[162]
ZFAS1	↑	LIN28 CAPRIN1	Enhances invasion by targeting LIN28 and CAPRIN1.	[163]
**circ RNAs**				
circ-RanGAP1	↑	VAGFA	Increases GC invasion via miR-877-3p/VEGFA axis.	[153]
circBGN	↑	IL6/STAT3	Prompts invasion through miR-149-5p/IL6.	[164]
circDUSP6	↑	IVNS1ABP	Promotes invasion by sponging miR-145-5p.	[165]
hsa-circ-0052001	↑	MAPK signal pathway.	Facilitates invasion of GC cells through the MAPK signal pathway.	[166]
circNRIP1	↑	AKT1	Sponges miR-149-5p to increase the AKT1 expression.	[150]
circSHKBP1	↑	HUR VEGF HSP90	Exosomal circSHKBP1 modulates the miR-582-3p/HUR/VEGF pathway and inhibits HSP90 degradation.	[167]
circUBE2Q2	↑	STAT3	Promotes GC invasion via the circUBE2Q2-miR-370-3p-STAT3 axis.	[168]
circ-CEP85L	↓	NFKBIA	Increases NFKBIA expression by targeting miR-942-5p and impairs invasion.	[157]
hsa_circ_0005230	↑	RHOT1	Induces the EMT by regulating RHOT1 expression via sponging miR-1299, thus promoting invasion.	[169]
circPDIA4	↑	MAPK	Enhances invasion and metastasis by bounding the ERK/MAPK pathway.	[151]
**ncRNA inhibiting invasion**				
**miRNA**				
miR-9	↓	cyclin D1, Ets1	Represses invasion and metastasis of GC.	[170]
miR-140-5p	↓	WNT1 YES	Inhibits cell invasion via Wnt/β-catenin signaling.	[171]
miR-181a	↑	Caprin-1	Inhibits caprin-1 and enhances GC invasion.	[172]
miR-181-5p	↓	ZFP91.	Serves as a tumor suppressor by targeting ZFP91.	[173]
miR-200b-3p	↓	CXCL12 CXCR7	Suppresses invasion by modulating the CXCL12/CXCR7 axis.	[155]
miR-214	↓	FGF9	Suppressed invasion via targeting FGF9 in CAFs and regulating the EMT.	[156]
miR-330-3p	↑	PRRX1	Repressed EMT and invasion via inhibiting PRRX1-regulated Wnt/β-catenin signaling.	[174]
miR-505	↓	HMGB1	Suppresses invasion by targeting HMGB1.	[175]
miR-520a-3p	↑	SKA2	Suppresses invasion using invasion assay.	[176]
miR-1254	↓	Smurf1	Suppresses invasion and EMT, and reduces the PI3K/AKT signaling via downregulating Smurf1.	[177]
miR-3648	↓	FRAT1, FRAT2	Suppresses invasion by targeting FRAT-1 and-2 via the Wnt/β-catenin signaling.	[154]
miR-3664-5p	↓	MTDH	Inhibits invasion through the NF-κB signaling by targeting MTDH.	[178]
**lnc RNAs**				
CA3-AS1	↓	BTG3	Increases the expression of BTG3 and inhibits invasion by targeting miR-93-5p.	[179]
LINC00473	↑	MMP2, MMP9	Restores invasion via regulating MMP2 and MMP9 expression.	[149]
**circ RNAs**				
circ-ITCH	↓	Wnt/β-catenin pathway	Suppresses by sponging miR-17via Wnt/β-catenin pathway.	[180]
circRELL1	↓	EPHB3	Represses invasion via the circRELL1/miR-637/EPHB3 axis.	[181]
circ_0026344	↓	PDCD4	Increases PDCD4 expression by targeting miR-590-5p, thus impairing invasion.	[182]

**Table 2 cancers-16-00054-t002:** The summary of factors affecting the invasion of gastric cancer (excluding ncRNA). cnRNA = non-coding RNA, EMT = epithelial–mesenchymal transition, GC = gastric cancer, Treg = regulatory T cell, PAR1 = Proteinase-activated receptor1, RhoGDI2 = Rho GDP differentiation inhibitor 2, ADMA = asymmetric dimethylarginine, HIF-1α = hypoxia-inducible factor-1α, NF-κB = nuclear factor-kappa B, PI3K = phosphatidylinositol-3 kinase, mTOR = mammalian target of rapamycin, FAK = focal adhesion kinase, BAG4 = Bcl2-associated athanogene 4, CEACAM6 = Carcinoembryonic antigen-related cell adhesion molecule 6, CSNK2 = casein kinase II, STAT3 = signal transducer and activator of transcription 3, GDF15 = growth differentiation factor 15, S100A16 = S100 calcium-binding protein A16, CRYABZ = Alpha B-crystallin, CHIP = carboxyl terminus of Hsc-70-interacting protein, ZBTB20 = Zinc finger and BTB domain containing 20, TGF-β = Transforming growth factor-β, TRPV2 = transient receptor potential vanilloid 2, DACH1 = Dachshund homolog 1, TAF15 = TATA-box-binding protein-associated Factor 15, MAGI1 = membrane-associated guanylate kinase inverted 1, MMP = matrix metalloproteinase, uPA = urokinase-type plasminogen activator, uPAR = uPA receptor, CAF = cancer-associated fibroblast, TME = tumor microenvironment, BM = basement membrane, FAP = fibroblast activation protein, IQGAP1 = IQ motif-containing GTPase-activating protein 1, ATP5B = adenosine triphosphate synthase F1 β subunit, ↓ = downregulation; ↑ = upregulation, N.A. = not applicable.

Classification	Specific Elements	Alterations	Comments	Ref
**Invasion**				
**phenotype**				
**EMT**	Collagen type IVα1	↑	Prompts EMT and the invasion ability of GC cells via the Hedgehog signaling pathway.	[25]
**MAT**	ROCK	↑	ROCK activities induce MAT in scirrhous GC.	[31]
**Signaling pathway**				
**Rho/ROCK signaling**	PAR1	↑	PAR-1 by thrombin prompts GC cell invasion by the acquisition of morphological change by targeting RhoA and Rac1.	[48]
	RhoJ	↑	Relates to the EMT procedure to increase the invasion via IL-6/STAT3.	[49]
	G17E (RhoA mutant)	↑	Upregulates Vav1 expression and facilitates the invasion via MMP-9.	[50]
	RhoGDI2	↑	Prompts Rac1 activity and enhances the binding between Rac1 and Filamin A, which leads to increased invasion.	[52]
**Wnt/β-catenin signaling**	CCT5	↑	Enhances GC cell invasion and lymph node metastasis by activating Wnt/β-catenin signaling activity and EMT.	[54]
	Capn4	↑	Induces invasion by promoting MMP9 expression via Wnt/β-catenin signaling.	[55]
	ADMA	↑	Promotes the expression of β-catenin and activates the Wnt/β-catenin pathway, thereby enhancing invasion and EMT.	[56]
	HIF-1α	↑	HIF-1α regulates the Wnt/β-catenin pathway, activates uPA and MMP-7 expression, and contributes to the enhanced invasion.	[57]
**PI3K/AKT/mTOR signaling**	PTEN	↓	Represses the PI3K/NF-κB pathway, which leads to the inhibition of invasion.	[60]
	BAG4	↑	Activates the PI3K/AKT/NF-κB/ZEB1 axis and enhances the invasion and metastasis.	[61]
	CEACAM6	↑	Induces EMT and promotes the production of MMP-9 through PI3K/AKT pathway.	[63]
	CSNK2	↑	Acts as an oncogene in invasion via EMT and the PI3K/Akt/mTOR signaling.	[64]
**JAK/STAT signaling**	gastrokine2	↓	Restores the invasion by suppressing JAK2/STAT3 signaling and downregulated MMP-2 and MMP-9 activity.	[68]
	GDF15	↑	Promotes the invasion and EMT via STAT3 activation in refractory GC cells.	[69]
	leptin	↑	Activates the invasion by targeting the MEK and JAK-STAT pathways, which led to the preservation of stemness.	[70]
**NF-κB signaling**	ADAMTS16	↑	Enhances the invasion by targeting IFI27 via the NF-κB pathway.	[72]
	ADAMTS19	↓	Restores cell invasion by targeting S100A16 through the NF-κB pathway.	[73]
	CRYABZ	↑	Promotes invasion by overexpression of CRYABZ via the NF-κB-regulated EMT.	[74]
	CHIP	↓	Restores invasion through NF-κB subunits, RelA/p65, and RelB signaling by reducing TRAF2 activity.	[75]
	ZBTB20	↑	Overexpresses in GC cells by *Helicobacter pylori* activity and promotes invasion through MMP-2/-9.	[81]
**TGF-β signaling**	TRPV2	↑	Enhances invasion through the TGF-β signaling pathway.	[84]
	S100A4	↑	Enhances invasion by activation of the TGF-β/Smad signaling-mediated EMT.	[85]
	inhibin βA (TGF-β superfamily)	↑	Enhances invasion by activating the TGF-β signaling pathway.	[86]
	NNMT	↑	Increased the EMT markers and invasion via TGF-β signaling pathway.	[92]
	*DACH1*	↓	Impairs GC invasion and EMT by suppressing TGF-β signaling.	[93]
**RAS/RAF/ERK/MAPK signaling**	SIRT2	↑	Enhances invasion through RAS/ERK/JNK/MMP-9.	[94]
	TAF15	↑	Promotes the migration and invasion through RAF1/MEK/ERK signaling	[95]
	MAGI1	↓	Impairs invasion by affecting the expression of EMT-related molecules and MMP.	[96]
**Hippo pathway**	GNB4,	↑	Prompts invasion through the Hippo-YAP1 pathway by *Helicobacter pylori* infection	[102]
	FBXW5	↓	Increases invasion by downregulating the Hippo signaling.	[103]
**Proteases**				
**MMP**	MMP-1	↑	MMP-1 expression at the leading edge of invasive tumors can predict lymph node metastasis.	[110]
	MMP-2	↑	The expression of MMP2 was correlated with invasion, metastasis, and microvessel density in GC	[112]
	MMP-7	↑	Promotes levels of activator protein 1, gastrin which is closely associated with *H. pylori* infection.	[120]
	MMP-9	↑	The level of MMP9 in serum enhanced progressively depending on the depth of tissue invasion.	[116]
	MT1-MMP	↑	Promotes invasion of cells by modulating the vimentin and E-cadherin expression	[122]
**uPA**	uPAR	↑	uPAR modulates the proteolytic activity of the ECM to activate the plasminogen activation system.	[123]
**Chemokines**	CXCL12	↑	CXCL12 derived from CAFs enhances invasion via stimulating the β1-integrin clustering	[127]
	CXCL5	↑	Induces EMT and pre-cancerous activation of neutrophils, which lead to facilitate invasion.	[131]
	CXCL10/CXCR3	↑	Enhances invasion by stimulating MMP-2/9 secretion through PI3K/AKT pathway.	[132]
**Inflammatory cytokines**	TNF-α and IL-1β	↑	Stimulates gastric cell MMP-1, 13 secretions through RAS/RAF/ERK/MAPK signaling.	[97]
	IL-33	↑	Facilitates invasion and EMT triggered by CAFs by the ERK1/2-SP1-ZEB2 signaling through ST2L.	[133]
	IL-23	↑	IL-23 connecting to its receptor and thereby induced microtubules via the STAT3 signaling.	[134]
**Genes**	*DDOST, GNS, NEDD8, LOC51096, CCT3, CCT5, PPP2R1B*, and two ESTs.	↑	Nine of the 12 genes are relatively upregulated and three are downregulated in GC patients with lymph node metastasis.	[38]
	*UBQLN1, AIM2*, *USP9X*	↓		
	*CDH17* and *APOE*	N.A.	Correlates with invasion depth of tumors by serial analysis of gene expression (SAGE)	[40]
	*IQGAP1*	↑	Promotes cell invasion by targeting RhoC GTPase.	[42]
	*IQGAP3*	N. A.	IQGAP3 acts as an essential mediator of invasion and EMT through TGF-beta signaling.	[43]
	*ZYX*	↑	Regulates EMT via the WNK1/SNAI1 signaling to increase invasion.	[44]
	*DPP4*, *OLFM4*, *CLCA1*, *SI*, *MEP1A*.	N.A.	These genes were enriched in protein digestion and absorption and carbohydrate digestion pathways.	[45]
**Cells**	CAFs	N.A.	CAF-mediated TME remodeling promotes EMT and invasion, which is facilitated by the collagen-rich matrix.	[25]
	CAFs	N.A.	CAFs may produce gaps in the BM and stromal components which are bound to cell–cell junctions to prompt collective cell invasion	[136]
	FAP (CAF marker)	↑	Stromal FAP secreted from CAFs facilitates invasion through EMT by targeting the Wnt/β-catenin pathway.	[210]
**Others**	ATP5B	↑	ATP5B in the TME contributes to tumor invasion of GC via FAK/AKT/MMP2 pathway.	[114]
	NDRG1	↑	Enhances invasion cells by targeting MMP-9.	[117]
	semaphorin 5A	↑	Promotes invasion by increasing uPA expression through the PI3K/Akt signaling pathway.	[125]

## Data Availability

The data can be shared up on request.

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
