# Peer review of "Molecular Insight into Gastric Cancer Invasion—Current Status and Future Directions"

_cancers, 2023, doi:10.3390/cancers16010054_

Round 1

Reviewer 1 Report

Comments and Suggestions for Authors

The authors are summarising the current knowledge on mutations and molecular pathways involved in GC invasion. The authors start by introducing the concept of invasion plasticity and then describe mutations, molecular pathways and regulatory mechanisms shown to be involved in GC invasion. They also describe some interactions with the tumour microenvironment that were shown to modulate GC invasion. In general, the document has a well-defined structure and extensively covers up to date findings in the field.

However, I would suggest to further expand the following general aspects:

1.     In chapter 4 the authors nicely introduce the concept of invasion plasticity and even how it could mediate resistance to treatment. However, as they move into describing the different pathways a clear link to how these molecular pathways contribute to this plasticity and whether they are doing it in response to treatment is sometime missing.

2.     In some cases, when citing a pathway or molecule, the authors describe it as “involved” in invasion, without specifying whether it does promote or inhibit invasion, or whether it promotes transition from one type of invasion to another.

Representative examples among others:

·      Row 299 – “Phosphorylated PI3K/Akt/mTOR signalling affects many cellular biological activities, such as cell growth, differentiation, intracellular trafficking, and invasion.” In which direction?

·      Row 394 - ….”mediate the differentiation, growth, apoptosis, and invasion of cancer cells”. Does the Hippo pathway promote or inhibit them? In row 397 is says that the disruption of Hippo signalling is associated with GC invasion. Please clarify.

3.     Some molecules are described as promoting invasion “by”, “via”, or “through” other pathways; however, no details on how this is happening at the molecular level is provided.

Representative examples among others:

·      Row 253 - Rac signals can prompt AMT via the SCAR/WAVE2 conjugate for the plasticity of invasion modes

·      Row 258 - activation of PAR-1 by thrombin mediates GC cell invasion

·      Row 363 - TRPV2……enhanced GC cell invasion through the TGFß signalling pathway…

·      Row 365 - S100A4 enhances the invasion ability of GC cells by TGF-β activation

·      Row 400 - …GNB4 ….. prompted the invasion of GC through the Hippo-YAP1 pathway….

·      Row 476 – Stromal FAP secreted from CAFs facilitates GC cell invasion through EMT phenotype by targeting the Wnt/β-catenin pathway.

4.     Concerning chapter 4, how the authors comment the different signatures found in different studies? Also, most of the studies described are correlation studies, was the role of some of these genes functionally validated in the contest of GC? Are the genes listed in this chapter known to modulate the pathways described in the following chapters?

5.     A paragraph on the role/involvement of H. pylori should be added. H. pylori is cited in several chapters but never introduced in the clinical contest of GC.

6.     Chapter 7. The authors should introduce what NC-RNAs are, the different types of NC-RNAs and their different functions. Also, I would suggest dividing the NC-RNAs cited in this chapter into two separate sub-paragraphs: one describing NC-RNAs promoting GC invasion, and one describing those inhibiting GC invasion. It would be easier to follow.

7.     It is not clear the difference between Chapters 8 and 9. Both describe possible approaches to target GC invasion and some targets are repeated in the two chapters. My impression is that from row 633 to row 673 should be moved into chapter 8. In chapter 9 I would limit the discussion to the importance of targeting invasion plasticity and possible ways to do it.

Additional points that require the authors’ attention:

·      Row 288 - the following sentence seams incomplete: “Transfer extracellular signalling to the nucleus through receptors of cell surface and cytoplasmic moderators.”

·      Fig. 3 legend - This legend requires improvement. The authors could add a general description of what is represented in the figure. Is the list of acronyms meant to go here or in a dedicated section (e.g. abbreviations)?

·      Row 264 - The RhoA mutant G17E was not introduced. Why is this mutation relevant?

·      Row 277 - involving or including cancer?

·      Row 281 - ….”CCT5 attaches (???) E-cadherin….”

·      Row 326 - ….” restored (or inhibited?) the invasion of GC cells….”

·      Row 375 - “The RAS/RAF/ERK/MAPK signalling pathway is the most common signalling in the pathology of cancer”. What do the authors mean with “most common”? Most frequently activated? Mutated?

·      Row 393 – What do you mean with “progression” of tissue organs?

·      Row 415-417 – T, B, dendritic, and myeloid-derived suppressor cells are all immune cells

·      Row 421 - ….” the capacity of malignant cells to invade needs the degradation of the ECM by respective ECM remodelling issues.” The authors previously stated (Overview of invasion and metastasis row 125) that amoeboid invasion does not require degradation of the ECM. I would specify the type on invasion described in row 421.

·      Row 422 – It is not clear what the following is: “respective ECM remodelling issues”.

·      Row 488 – “…recent evidence suggests that perineural invasion (PNI) has been related to a substantial pathological process of various tumours in GCs.” This should be moved in the introduction where the different types of invasions are described and it should be expanded.

·      Row 492 – what is a “latent mediator”?

·      Row 498 – miR-214 or miR3648?

·      Row 521-524 – It is not clear whether circPDIA4 promotes or inhibits invasion.

·      Row 598 – It should be “9” not “4”.

·      Row 688 – restoration or inhibition?

·      Row 706-709 – Should this be removed?

Author Response

Response to reviewers

Reviewer #1:

  1. In chapter 4 the authors nicely introduce the concept of invasion plasticity and even how it could mediate resistance to treatment. However, as they move into describing the different pathways a clear link to how these molecular pathways contribute to this plasticity and whether they are doing it in response to treatment is sometime missing.

Response: We appreciate the reviewer’s comments. In compliance with the reviewer’s suggestion, we added the sentences indicating the relation of signaling with plasticity and resistance to treatment on rows 336-343, 367-369, 407-413, and 465-470.

  1. In some cases, when citing a pathway or molecule, the authors describe it as “involved” in invasion, without specifying whether it does promote or inhibit invasion, or whether it promotes transition from one type of invasion to another.

Representative examples among others:

  • Row 299 – “Phosphorylated PI3K/Akt/mTOR signalling affects many cellular biological activities, such as cell growth, differentiation, intracellular trafficking, and invasion.” In which direction?
  • Row 394 - ….”mediate the differentiation, growth, apoptosis, and invasion of cancer cells”. Does the Hippo pathway promote or inhibit them? In row 397 is says that the disruption of Hippo signalling is associated with GC invasion. Please clarify.

Response: We appreciate the reviewer’s comments. We have checked the manuscript carefully and rewrote it to clarify whether it promotes or inhibits invasion as possible.

  1. Some molecules are described as promoting invasion “by”, “via”, or “through” other pathways; however, no details on how this is happening at the molecular level is provided

Representative examples among others:

  • Row 253 - Rac signals can prompt AMT via the SCAR/WAVE2 conjugate for the plasticity of invasion modes
  • Row 258 - activation of PAR-1 by thrombin mediates GC cell invasion
  • Row 363 - TRPV2……enhanced GC cell invasion through the TGFß signalling pathway…
  • Row 365 - S100A4 enhances the invasion ability of GC cells by TGF-β activation
  • Row 400 - …GNB4 ….. prompted the invasion of GC through the Hippo-YAP1 pathway….
  • Row 476 – Stromal FAP secreted from CAFs facilitates GC cell invasion through EMT phenotype by targeting the Wnt/β-catenin pathway.

Response: We appreciate the reviewer’s comments. According to the reviewer’s suggestion, we have checked the manuscript and rewrote it in detail as possible.

  1. Concerning chapter 4, how the authors comment the different signatures found in different studies? Also, most of the studies described are correlation studies, was the role of some of these genes functionally validated in the contest of GC? Are the genes listed in this chapter known to modulate the pathways described in the following chapters.

Response: We appreciate the reviewer’s comments. In compliance with the reviewer’s suggestion, we added sentences on rows 235-236, and 263-269.

  1. A paragraph on the role/involvement of H. pylori should be added. H. pylori is cited in several chapters but never introduced in the clinical contest of GC.

Response: According to the reviewer’s suggestion, we added a paragraph on the role of H. pylori in the text on rows 413-423.

  1. Chapter 7. The authors should introduce what NC-RNAs are, the different types of NC-RNAs and their different functions. Also, I would suggest dividing the NC-RNAs cited in this chapter into two separate sub-paragraphs: one describing NC-RNAs promoting GC invasion, and one describing those inhibiting GC invasion. It would be easier to follow.

Response: We have added sentences about what nc-RNAs are on rows 574-579, and made the changes as suggested by the reviewer to subdivide chapter 7 and refer to them in Table 1. Appropriate changes have been made in the text.

  1. It is not clear the difference between Chapters 8 and 9. Both describe possible approaches to target GC invasion and some targets are repeated in the two chapters. My impression is that from row 633 to row 673 should be moved into chapter 8. In chapter 9 I would limit the discussion to the importance of targeting invasion plasticity and possible ways to do it.

Response: We appreciate the reviewer’s comments. According to the reviewer’s suggestion, we moved from row 633 to row 673 into chapter 8 to limit the discussion to the importance of targeting invasion plasticity.

Additional points that require the authors’ attention:

  • Row 288 - the following sentence seams incomplete: “Transfer extracellular signalling to the nucleus through receptors of cell surface and cytoplasmic moderators.”

→We changed these words to “Transfer extracellular signaling to the nucleus through receptors of cell surface and cytoplasmic moderators represents a central process for the development of malignancies.”

  • Fig. 3 legend - This legend requires improvement. The authors could add a general description of what is represented in the figure. Is the list of acronyms meant to go here or in a dedicated section (e.g. abbreviations)?

→According to the reviewer’s suggestion, we added a general description in this figure and deleted unnecessary acronyms.

  • Row 264 - The RhoA mutant G17E was not introduced. Why is this mutation relevant?

→We introduced G17E as: “G17E is a malignant phenotype resulting from RhoA mutations where the frequency is high in GC.” Appropriate change has been made in the text on rows 303-305.

  • Row 277 - involving or including cancer?

→We rewrote it to “including cancer”.

  • Row 281 - ….”CCT5 attaches (???) E-cadherin….”

→We rewrote it to “CCT5 binds to E-cadherin….”.

  • Row 326 - ….” restored (or inhibited?) the invasion of GC cells….”

→We rewrote it to “inhibited the invasion of GC cells…”.

  • Row 375 - “The RAS/RAF/ERK/MAPK signalling pathway is the most common signalling in the pathology of cancer”. What do the authors mean with “most common”? Most frequently activated? Mutated?

→We rewrote it to “The RAS/RAF/ERK/MAPK signaling pathway is one of the most familiar signaling in the pathology of cancer”

  • Row 393 – What do you mean with “progression” of tissue organs?

→We rewrote it to “development”.

  • Row 415-417 – T, B, dendritic, and myeloid-derived suppressor cells are all immune cells

→We rewrote it to “immune cells (T cells, B cells, neuroendocrine cells, dendritic cells (DCs), myeloid-derived suppressor cells (MDSCs))”.

  • Row 421 - ….” the capacity of malignant cells to invade needs the degradation of the ECM by respective ECM remodelling issues.” The authors previously stated (Overview of invasion and metastasis row 125) that amoeboid invasion does not require degradation of the ECM. I would specify the type on invasion described in row 421.
  • Row 422 – It is not clear what the following is: “respective ECM remodelling issues”.

→We rewrote this sentence to “For cancer cells, the capacity of malignant cells to invade needs the degradation of the ECM which is involved in ECM remodeling in collective and mesenchymal cell invasion mode”.

  • Row 488 – “…recent evidence suggests that perineural invasion (PNI) has been related to a substantial pathological process of various tumours in GCs.” This should be moved in the introduction where the different types of invasions are described and it should be expanded.

→According to the reviewer’s recommendation, we moved this sentence to Chapter 3 and added the explanation of PNI on rows 111-114.

  • Row 492 – what is a “latent mediator”?

→We changed “latent” with “important”.

  • Row 498 – miR-214 or miR3648?

→We apologize we did careless mistakes. We changed “miR-214” with “miR-3648”.

  • Row 521-524 – It is not clear whether circPDIA4 promotes or inhibits invasion.

→We added the words “prompted GC invasion” on row 612.

  • Row 598 – It should be “9” not “4”.

→We apologize we did careless mistakes. We changed “4” with “9”.

  • Row 688 – restoration or inhibition?

→We changed “restoration” with “inhibition”.

  • Row 706-709 – Should this be removed?

→We apologize we did careless mistakes. We deleted these sentences.

Reviewer 2 Report

Comments and Suggestions for Authors

The manuscript “Molecular insight into gastric cancer invasion– current status and future directions” is a review regarding the available literature on molecular mechanism of gastric cancer invasion and metastasis, as well as possible implications for identifying therapeutic targets.

The manuscript may be of interest for the readers, and in general is well written. However, there are some important concerns. Authors are requested to address the following concerns since, as it is, the manuscript cannot be accepted for publication.

Major

1.       The manuscript seems to be written carelessly. There are some strange things. For instance, some paragraphs are written in gray font instead of black, or highlighted in yellow (e.g. figure 2 legend; line 235-236). The conclusion section starts with number 10, while the previous paragraph has number 4. These are signs of an inaccurate check of the manuscript before submission. The worst thing appears in line 706 “Authors should discuss the results and how they can be interpreted from the perspective of previous studies and of the working hypotheses. The findings and their implications should be discussed in the broadest context possible. Future research directions may also be highlighted.” Did the authors read again the whole manuscript after formatting it?!

2.       Weirdness is present also in the Institutional Review Board Statement. The paragraph hasn’t been edited.

3.       Table 1: Instead of writing up and down in the columns alterations, which is too simplistic, authors should replace them with arrows (the same in table 2). Moreover, the column of reference is sometimes in bold. 

4.       The manuscript lacks of important studies available in literature. A major player in gastric cancer progression is the enzyme nicotinamide N-methyltransferase (NNMT) which has been proven to be upregulated in gastric cancer (PMID: 17079861; PMID: 36139012), and whose expression is correlated to a worse prognosis (PMID: 27152242; PMID: 36977555) and correlated to EMT transition in gastric cancer (PMID: 29541230).  Finally, NNMT seems to be a master regulator of CAFs (PMID: 31043742).

A number of NNMT inhibitors are already available and could be tested for gastric cancer management (PMID: 34572571; PMID: 34704059; PMID: 34424711). All these considerations cannot be ignored in a review article regarding molecular mechanism of gastric cancer invasion, metastasis and targets.

5.       In table 2, underlining the category (e.g. Invasion Phenotype, Signaling pathway…” creates confusion and makes reading the table difficult. I suggest to change the style.

6.       What about the available literature of gastric cancer stem cells?

Comments on the Quality of English Language

Moderate editing of English language required

Author Response

Response to reviewers

Reviewer #2:

  1. The manuscript seems to be written carelessly. There are some strange things. For instance, some paragraphs are written in gray font instead of black, or highlighted in yellow (e.g. figure 2 legend; line 235-236). The conclusion section starts with number 10, while the previous paragraph has number 4. These are signs of an inaccurate check of the manuscript before submission. The worst thing appears in line 706 “Authors should discuss the results and how they can be interpreted from the perspective of previous studies and of the working hypotheses. The findings and their implications should be discussed in the broadest context possible. Future research directions may also be highlighted.” Did the authors read again the whole manuscript after formatting it?!

Response: We appreciate the reviewer's comments. We are very sorry for the trouble we caused due to our cursoriness. We have carefully checked the manuscript and revised it as the reviewer mentioned.

We changed the gray font to black and deleted the highlight. We rewrote “chapter 4” to “chapter 9” on row 749. I deleted sentences from the rows 806 to 809.

  1. Weirdness is present also in the Institutional Review Board Statement. The paragraph hasn’t been edited.

Response: We appreciate the reviewer's comments and apologize for the careless mistakes. We added “Institutional Review Board Statement” in the text.

  1. Table 1: Instead of writing up and down in the columns alterations, which is too simplistic, authors should replace them with arrows (the same in table 2). Moreover, the column of reference is sometimes in bold. 

Response: We appreciate the reviewer's comments and apologize we making careless mistakes. We changed “up” to “↑” and “down” to “↓”. We also revised the column of references in Table 1.

  1. The manuscript lacks of important studies available in literature. A major player in gastric cancer progression is the enzyme nicotinamide N-methyltransferase (NNMT) which has been proven to be upregulated in gastric cancer (PMID: 17079861; PMID: 36139012), and whose expression is correlated to a worse prognosis (PMID: 27152242; PMID: 36977555) and correlated to EMT transition in gastric cancer (PMID: 29541230).  Finally, NNMT seems to be a master regulator of CAFs (PMID:31043742).

A number of NNMT inhibitors are already available and could be tested for gastric cancer management (PMID: 34572571; PMID: 34704059; PMID: 34424711). All these considerations cannot be ignored in a review article regarding molecular mechanism of gastric cancer invasion, metastasis and targets.

Response: We appreciate the reviewer’s comments. As the reviewer recommended, we added a paragraph on the role of NNMT in the text on rows 435-444 and in Table 2.

  1. In table 2, underlining the category (e.g. Invasion Phenotype, Signaling pathway…” creates confusion and makes reading the table difficult. I suggest to change the style.

Response: We appreciate the reviewer’s comments. We changed the style of Table 2 as the reviewer recommended.

  1. What about the available literature of gastric cancer stem cells?

Response: We appreciate the reviewer’s comments. In compliance with the reviewer’s suggestion, we added the sentences on rows 465-470 and 198-207.

Reviewer 3 Report

Comments and Suggestions for Authors

see attachment

Author Response

Response to reviewers

Reviewer #3:

Specific comments

  1. in the abstract section, it is better that the authors give some their results in this study.

Response: We appreciate the reviewer’s comments. According to the reviewer’s suggestion, we added our results in this review in the abstract section.

  1. In the methods section, the authors mentioned that some keywords used for searching including “signal transduction and non-coding RNAs. Why did authors used these two keywords that they are not related directly to invasion.

Response: Thank you for the reviewer’s helpful comments. We deleted “signal transduction” and “non-coding RNAs” in the methods section.

  1. Considering that most novel findings present in research articles, but the authors mentioned that they used review articles for their search resulting to they miss novel finding about this issue.

Response: We appreciate the reviewer's comments. We mainly searched the literature in the latest research articles. Additionally, we also include a review and guidelines for explaining the concept and current status of issues regarding GC invasion. We added these comments in the Methods section on rows 82-85 of the revised manuscript.

  1. It is better that authors give the numbers of articles that have their criteria and used in this study.

Response: We appreciate the reviewer's comments. We have added a comment regarding the number of literature in the method section on rows 90-92, as suggested by the reviewer.

Reviewer 4 Report

Comments and Suggestions for Authors

Comments:

1. Do the results presented in the latest review indicate any variations across different countries?

2. Given that Japan has the highest gastric cancer rate globally, could you provide additional information on how the Japanese have been combatting gastric cancer in recent decades?

3. Do the results presented in the latest review indicate any variations between intestinal type and diffuse type in gastric cancers?

4. What are the "chapter 3", "chapter 4", "chapter 6" on Figure 1?

5. As p53 stands out as the most prevalent mutation in gastric cancer, could you incorporate additional details regarding the role and implications of p53 mutations in gastric cancers?

6. Could you provide information and data pertaining to PDCD4 and other tumor suppressors in the context of gastric cancer?

7. What are the "chapter 5", "chapter 7" on Figure 3?

8. In Table 1, how is the fold change determined for distinguishing between upregulation and downregulation?

Author Response

Response to reviewers

Reviewer #4:

  1. Do the results presented in the latest review indicate any variations across different countries?

Response: We appreciate the reviewer’s comments. We have carefully checked the literatures whether there are variations across different countries. Unfortunately, we found no variation in the results of this review concerning the molecular mechanism of the invasion in GC, probably because the search results mainly consisted of basic research. The study comparing the GC exome profiles of 319 Asian patients to 212 non-Asian patients revealed the distinct GC subclass with alcohol-associated mutation signature and Asian-specific defective aldehyde dehydrogenase 2 family member allele by using whole exome sequencing (Suzuki et al. Sci Adv 2020 May 6;6(19): eaav9778). However, there is no description concerning invasion. We will keep this in mind and make the use of reviewer’s suggestion for the next article.

  1. Given that Japan has the highest gastric cancer rate globally, could you provide additional information on how the Japanese have been combatting gastric cancer in recent decades?

Response: We appreciate the reviewer’s comments. In compliance with the reviewer’s suggestion, we added the sentences on rows 34 and 37-49 in the revised manuscript.

  1. Do the results presented in the latest review indicate any variations between intestinal type and diffuse type in gastric cancers?

Response: We appreciate the reviewer’s comments. Diffuse-type GC comprises of poorly differentiated cells that become highly invasive compared with intestinal type. We modified the sentences as the reviewer pointed out on rows 52-57.

  1. What are the "chapter 3", "chapter 4", "chapter 6" on Figure 1?
  2. What are the "chapter 5", "chapter 7" on Figure 3?

Response: We are sorry for bothering the reviewer. We sought to represent how these chapters related to the invasion and metastatic process in Figures. We rewrote the sentence in the figure legend to clarify.

  1. As p53 stands out as the most prevalent mutation in gastric cancer, could you incorporate additional details regarding the role and implications of p53 mutations in gastric cancers?

Response: We appreciate the reviewer’s comments. We added the sentences concerning p53 mutations on rows 228-232 in the revised manuscript.

  1. Could you provide information and data pertaining to PDCD4 and other tumor suppressors in the context of gastric cancer?

Response: We appreciate the reviewer’s comments. According to the reviewer’s suggestion, we described sentences regarding PDCD4 and PTEN on rows 590-595, and in the revised text and table.

  1. In Table 1, how is the fold change determined for distinguishing between upregulation and downregulation?

Response: We appreciate the reviewer’s comments. NcRNAs upregulated in GC samples compared to normal controls are marked with ↑, while the downregulated ncRNAs are depicted with ↓.

Round 2

Reviewer 1 Report

Comments and Suggestions for Authors

I would like to thank the authors for addressing all my comments.

Reviewer 2 Report

Comments and Suggestions for Authors

The authors improved the manuscript addressing all the raised issued. Therefore, it can be accepted for publishing.

Comments on the Quality of English Language

Moderate editing of English language required

Reviewer 4 Report

Comments and Suggestions for Authors

No more comments